# Proteomic Analysis of 3T3-L1 Adipocytes Treated with Insulin and TNF-α

**DOI:** 10.3390/proteomes7040035

**Published:** 2019-10-20

**Authors:** Hayley Chan, Ketaki P. Bhide, Aditya Vaidyam, Victoria Hedrick, Tiago Jose Paschoal Sobreira, Thomas G. Sors, Ryan W. Grant, Uma K. Aryal

**Affiliations:** 1Department of Computer Science, Purdue University, West Lafayette, IN 47907, USA; hachan@iu.edu (H.C.); avaidyam@purdue.edu (A.V.); 2College of Agriculture, Purdue University, West Lafayette, IN 47907, USA; bhide@purdue.edu; 3Purdue Proteomics Facility, Bindley Bioscience Center, Purdue University, West Lafayette, IN 47907, USA; vhedrick@purdue.edu (V.H.); sobreira@purdue.edu (T.J.P.S.); 4Purdue Institute of Inflammation, Immunology and Infectious Disease, Purdue University, West Lafayette, IN 47907, USA; tsors@purdue.edu; 5Department of Nutrition Science, Purdue University, West Lafayette, IN 47907, USA; r.w.grant@gmail.com; 6Department of Comparative Pathobiology, College of Veterinary Medicine, Purdue University, West Lafayette, IN 47907, USA

**Keywords:** 3T3-L cells, insulin, tumor necrosis factor-α, type II diabetes, quantitative proteomics, mass spectrometry

## Abstract

Insulin resistance is an indication of early stage Type 2 diabetes (T2D). Insulin resistant adipose tissues contain higher levels of insulin than the physiological level, as well as higher amounts of intracellular tumor necrosis factor-α (TNF-α) and other cytokines. However, the mechanism of insulin resistance remains poorly understood. To better understand the roles played by insulin and TNF-α in insulin resistance, we performed proteomic analysis of differentiated 3T3-L1 adipocytes treated with insulin (Ins), TNF-α (TNF), and both (Ins + TNF). Out of the 693 proteins identified, the abundances of 78 proteins were significantly different (*p* < 0.05). Carnitine parmitoyltransferase-2 (CPT2), acetyl CoA carboxylase 1 (ACCAC-1), ethylmalonyl CoA decarboxylase (ECHD1), and methylmalonyl CoA isomerase (MCEE), enzymes required for fatty acid β-oxidation and respiratory electron transport, and β-glucuronidase, an enzyme responsible for the breakdown of complex carbohydrates, were down-regulated in all the treatment groups, compared to the control group. In contrast, superoxide dismutase 2 (SOD2), protein disulfide isomerase (PDI), and glutathione reductase, which are the proteins responsible for cytoskeletal structure, protein folding, degradation, and oxidative stress responses, were up-regulated. This suggests higher oxidative stress in cells treated with Ins, TNF, or both. We proposed a conceptual metabolic pathway impacted by the treatments and their possible link to insulin resistance or T2D.

## 1. Introduction

The alarmingly increasing frequency of obesity and non-insulin-dependent (type II) diabetes mellitus (T2D) in the United States poses a serious health crisis in the near future. To address this, we need to better understand the molecular factors and cellular pathways that are responsible for insulin resistance. Insulin is a dipeptide hormone secreted by pancreatic islet β cells, which is essential for maintaining normal blood glucose levels [1]. Insulin resistance is defined as the inability of a cell to maintain glucose homeostasis, respond to the physiological level of insulin [2,3], and is a characteristic condition of the early state of T2D. 

Insulin lowers blood glucose levels by regulating glucose metabolism via glycolysis, gluconeogenesis, and glycogenesis pathways [4]. Glucose is converted into pyruvate or lactate via glycolysis by sequential glycolytic enzymes, including bifunctional 6-phosphofructo-2-kinase/fructose-2,6-bisphosphatase 1 (PFKFB1) [5]. Insulin dephosphorylates PFKFB1 to activate kinase activity and promote glycolysis. Gluconeogeneis, on the other hand, generates glucose from pyruvate or lactate, and glycogenesis converts glucose into glycogen granules. Gluconeogenesis is regulated by rate-limiting phosphoenolpyruvate carboxykinase (PEPCK) and glucose-6-phosphatase (G6Pase) [6]. Insulin is known to reduce PEPCK and G6Pase expression through serine/threonine kinase AKT signaling pathways resulting in the supression of gluconeogenesis [7,8]. Glycogenesis, under other conditions, is regulated by the level of glycogen synthase (GS) and glycogen phosphorylase (GP) [9]. Insulin promotes glycogenesis by dephosphorylation and activation of GS or dephosphorylation and inactivation of GP via the AKT pathway [10,11]. In addition, glucokinase (GK) also plays a key role for glucose utilization in the liver through glucose phosphorylation [12]. Insulin up-regulates GK expression [13], and its activity is also regulated by the interaction with other regulatory proteins [14]. However, the mechanism(s) by which insulin selectively regulates glycolysis, gluconeogenesis, and glycogenesis are still unclear. 

Many factors can cause insulin resistance, including genetic factors in some ethnic populations, environmental factors, and behavioral factors, such as physical inactivity, over-nutrition, and obesity [15]. Obesity is known to be linked with an increased risk of insulin resistance through the release of increased amounts of non-esterified fatty acids, glycerol, hormones, and pro-inflammatory cytokines from adipose tissues [16]. The dysfunction of pancreatic islet β-cells, due to insulin resistance, leads to the failure of the body to control blood glucose levels [17]. Higher amounts of TNF-α and other cytokines have been reported in insulin-resistant adipocytes [16]. Insulin resistant individuals also contain an elevated level of circulating insulin to keep blood glucose levels under control [18]. Studies have shown higher levels of TNF-α and higher expression of TNF-α mRNA in the tissues of obese, insulin-resistant patients with T2D, as well as in the adipose tissues of several rodent models of obesity and T2D [19,20,21]. Thus, TNF-α, a pro-inflammatory cytokine, plays a central role in some types of insulin resistance [22]. Consequently, the conditions for insulin resistance can be modeled by treating cells with insulin or TNF-α [18,23]. Cellular metabolic pathways and mechanisms of TNF-α mediated insulin resistance are unclear, and in many cases, contradictory. For example, TNF-α treatment resulted in decreased expression of insulin-sensitive glucose transporter GLUT4 in cultured cells [24], but normal GLUT4 levels in insulin resistant rodents and human muscles [25,26]. The GLUT4 knockout mice are moderately insulin resistant [27] and TNF knockout mice respond more efficiently to an exogenous dose of insulin and glucose than wild-type [28,29].

In this study, we performed a quantitative proteomic analysis of differentiated 3T3-L1 adipocytes treated with Ins only, with TNF only, and both Ins + TNF, to identify proteins that are differentially expressed, due to the treatments. The 3T3-L1 adipocytes develop insulin resistance in response to insulin, TNF or saturated fatty acids, and transcriptional response to TNF begins within the first 1 h of the treatment, and changes in protein phosphorylation associated with insulin resistance can be detected as early as 6 h [30,31]. We identified 693 proteins of which 643 proteins were common to all experimental conditions and expression of 78 proteins was significantly different in one or more treatments compared to the control. Mitochondrial and cytosolic proteins involved in sequential fatty acid beta oxidation and carbohydrate metabolism were down-regulated, and those involved in folding, translation and oxidative stress responses were up-regulated in response to either, Ins only, TNF only or Ins + TNF treatments. 

## 2. Materials and Methods

### 2.1. Cell Culture

Murine 3T3-L1 pre-adipocytes, [32] were grown, maintained, and induced, in order to differentiate using a standard protocol [33]. Fully differentiated adipocytes were maintained in DMEM (Sigma-Aldrich, St Louis, MO, USA) supplemented with 10% FBS (Sigma-Aldrich) until two days before experimentation, when cells were fed with 10% calf serum (Sigma-Aldrich). Prior to treatments, media was changed to low-glucose (5.5 mM) DMEM (Sigma-Aldrich) and 1% calf serum overnight. Cells were treated with 5 nM TNF-α (R&D Systems) for 8 h (TNF), 30 min with 10 nM insulin either alone (Ins) or followed by TNF treatment (Ins + TNF). 

### 2.2. Cell Lysis and Protein Extraction 

Prior to harvesting, cells were thoroughly rinsed at 37 °C. Differentiated 3T3-L1 cells were pelleted by centrifugation (Centrifuge 5810 R, Eppendorf, Hauppauge, NY, USA) at 14,000 rpm for 5 min at 4 °C. After washing cell pellets 3× with 200 µL of 20 mM PBS buffer, pH 7.5, the cells were suspended in 150 µL of wash buffer (20 mM PBS buffer, pH 7.5) and transferred to 150 µL size Barocycler microtubes. Freshly prepared serine protease inhibitor phenylmethylsulfonyl fluoride (PMSF) was added to each tube to achieve the final concentration of 1 mM, sealed and lysed in the Barocyler (Pressure Biosciences Inc., South Easton, MA, USA) at 35,000 psi for 60 min. The Barocycler was operated at 4 °C, holding 50 s at 35,000 psi followed by 10 s at atmospheric pressure per cycle for a total of 60 cycles or 1 h. Cell lysate was centrifuged at 14,000 rpm for 15 min at 4 °C and proteins in the supernatant were precipitated by adding 4 volume of cold (−20 °C) acetone and incubating overnight at −20 °C. After centrifugation at 14,000 rpm for 15 min at 4 °C, pellets were dried in a vacuum centrifuge (~1 min with heating system on), and dried protein pellets were dissolved in 40 µL of 8 M urea by incubating at room temperature for 1 h with continuous vortexing. The protein concentration was measured using a bicinchoninic acid (BCA) assay using bovine serum albumin (BSA) as a standard (Pierce Chemical Co., Rockford, IL, USA). 

### 2.3. Trypsin/LysC Digestion

Samples containing 50 µg of total protein were transferred to a new tube for reduction, alkylation, and proteolysis as described previously [34,35]. Briefly, proteins were reduced in 10 mM dithiothreitol (DTT), alkylated in 20 mM iodoacetamide (IAA), and digested in the Barocycler using a mass spec grade trypsin and Lys-C mixture from Promega at a 1:25 (*w*/*w*) enzyme-to-substrate ratio. Digestions were performed at 50 °C and 20,000 psi for 1 h (60 cycles; 50 s at 20,000 psi and 10 s at atmospheric pressure. The digested peptides were desalted using Pierce C18 spin columns (Pierce Biotechnology, Rockford, IL, USA). Peptides were eluted using 80% acetonitrile (ACN) containing 0.1% formic Acid (FA) and dried in a vacuum centrifuge at room temperature. Dried peptides were re-suspended in 80 µL of the buffer containing 97% water, 3% ACN and 0.1% FA. Peptide concentration was measured again using BCA assay. The volume of each peptide sample was adjusted to 0.2 µg/µL and 5 μL (1 µg of the total peptides) was used for LC-MS/MS analysis. 

### 2.4. Mass Spectrometry Analysis

Samples were analyzed by reverse-phase HPLC-ESI-MS/MS using a Dionex UltiMate 3000 RSLC Nano System (Thermo Fisher Scientific, Waltham, MA, USA), which was coupled to a Q-Exactive™ HF Hybrid Quadrupole-Orbitrap MS (Thermo Fisher Scientific) and a Nanospray Flex™ Ion Source (Thermo Fisher Scientific). Peptides were loaded to the trap column (300 μm ID × 5 mm) packed with 5 μm 100 Å C18 PepMap™ medium and then separated on an Acclaim™ PepMap™ 100 C18 analytical column (75 μm ID × 15 cm) packed with 2 μm 100 Å PepMap C18 medium (Thermo Fisher Scientific). 

The peptides were separated using a 120 min gradient method. Mobile phase solvent A was 0.1% FA in water and solvent B was 0.1% FA in 80% ACN. Peptides were loaded to the trap column in 100% buffer A for 5 minutes, and eluted with a linear 80 min gradient of 5 to 30% buffer B and reaching 45% B in 91 min, 100% of B in 93 min at which point the gradient was held for 7 min before reverting back to 5% B at 100 min. The column was equilibrated at 5% of B for 20 min. The samples were loaded at a flow rate of 5 μL/min for 5 min and eluted from the analytical column at a flow rate of 300 nL/min. The mass spectrometer was operated using standard data-dependent mode acquiring MS/MS for the top 20 precursors. The full scan MS spectra were collected in the 400–1600 *m/z* range with a maximum injection time of 100 milliseconds at a resolution of 120,000 (at 200 *m/z*). Fragmentation of precursor ions was performed by high-energy C-trap dissociation (HCD) with the normalized collision energy of 27 eV. MS/MS scans were acquired at a resolution of 15,000 (at *m/z* 200). The dynamic exclusion was set to 15 s to avoid repeated scanning of identical peptides.

### 2.5. Data Analysis

LC-MS/MS data were searched using MaxQuant software (v. 1.5.3.28) [36,37,38] for protein identification and label free MS1 quantitation. The MS/MS spectra were searched against the Uniprot mouse database (downloaded on 02/15/2018). The minimal length of 6 amino acids was required in the database search. The database search was performed with the precursor mass tolerance set to 10 ppm and MS/MS fragment ions tolerance was set to 20 ppm. Database searches were performed with enzyme specificity for trypsin and Lys-C, allowing up to two missed cleavages. The oxidation of methionine was defined as a variable modification, and carbamidomethylation of cysteine was defined as a fixed modification. The ‘unique plus razor peptides’ were used for peptide quantitation. Razor peptides are those assigned to the protein group with the highest number of peptides identified. The false discovery rate (FDR) of peptides and proteins identification was set at 1%. The results were filtered to remove contaminants and also those identified as reverse hit. Similarly, proteins with LFQ ≠ 0 and MS/MS ≥ 2 in at least two replicates were retained for further analysis.

### 2.6. Gene Ontology (GO), Statistical, and Cluster Analysis

GO terms for the cellular component and molecular function were mapped using the Panther Gene Ontology Consortium Slim Cellular Component and Biological Processes analysis. Bonferroni analysis was used and only significantly enriched genes were included (*p* < 0.05).

The statistical analysis of LC-MS/MS search results was conducted using the InfernoRDN software (previously known as DAnTE) [39] that uses an R script for the statistical tests. Differentially expressed proteins were identified by two-way analysis of variance (ANOVA) using protein LFQ values and the proteins with a *p*-value ≤ 0.05 were considered for further analysis and biological interpretation. Classification for the molecular function category is based on gene ontology (GO).

## 3. Result and Discussion

### 3.1. Experimental Overview and LC-MS Reproducibility

Figure 1 shows experimental overview. Cells were treated with either Ins, TNF and both Ins + TNF. LC-MS/MS data were acquired on a Thermo Q-Exactive Orbitrap HF MS using three biological replicates. LC-MS reproducibility was tested running 3 technical replicates. The average coefficient of variation (CV) of peptide intensities was 17.2% and median CV was 15.8% (Appendix A), indicating good quantitative reproducibility. The correlation co-efficient (r^2^) of peptides intensity was 0.945 between replicate 1 and 2, and 0.987 between replicate 1 and 3 (Appendix A). The r^2^ of protein intensities was 0.997 between replicate 1 and 2, and 0.998 between replicate 1 and 3 (data not shown). Accuracy of LC-MS/MS analysis was also evaluated using 3 biological replicates, and showed strong correlation as shown in Figure 2A for the control samples. These results were consistent for other biological replicates as well. 

### 3.2. Analysis of Differentially Expressed Proteins

We identified a total of 5022 peptides (Appendix A) mapped to 1050 proteins in at least one of the 12 samples (3 replicates for each of the 4 condition). We filtered these proteins to remove those that have no quantifiable MS1 peak intensity, and 0 or 1 MS/MS count. We then only considered proteins that were identified in at least 2 of the 3 biological replicates for downstream statistical analysis. After applying these filtering steps, we have a final list of 693 proteins (Appendix A). Functional classification of these proteins showed that the list was diverse including kinases, transcription and translational factors, transporters, carbohydrate and sugar metabolism, structural proteins, and enzymes (Appendix A). These proteins belong to diverse cellular compartments, including mitochondrion, nucleus, cytosol, extracellular exosomes, integral component of membrane, plasma membrane, and ER (Appendix A). Using a threshold of *p* value ≤ 0.05, we identified 78 proteins that were significantly different in cells treated with Ins only, TNF only, or both, compared to the control (Table 1). Most quantitative proteomics methods use fold changes to determine differentially expressed proteins, and a 2-fold increase or decrease is the most commonly applied threshold. However, a fold change to determine altered proteins is confounded by biases in protein abundances [40]. For example, a 1.5 or 2-fold increase can be highly stringent for highly abundant proteins, but for low abundance proteins, this level of change might simply represent technical noise. Therefore, rather than relying on fold changes, we used the one-way ANOVA test to determine differentially expressed proteins and used *p* values of ≤ 0.05 as a cut-off to determine significantly altered protein expression between the treatments. We found that many significantly different proteins across experimental groups have fold changes ≤1.5 (Appendix A). Of the 78 significant proteins, the expression of 18 proteins was at least 2-fold higher and the expression of 28 proteins was at least 2-fold lower in at least one of the 3 treatment groups (Appendix A). Changes in expression of 32 significantly different proteins was less than 2-fold and 24 proteins were uniquely identified in Ins + TNF (Figure 2B and Appendix A). Among the 78 differentially expressed proteins, the expression of 20 proteins was up and 30 proteins was down in all the treatments compared to the control (Table 1). Eight (8) proteins were up and 4 proteins were down in Ins only treatment, 3 proteins were up and 5 proteins were down in TNF only, and 4 proteins were up and 4 proteins were down in Ins + TNF (Table 1). The expression of 7 proteins was detected only in the control, 5 proteins only in Ins, 4 proteins only in TNF, and 15 proteins only in Ins + TNF (Figure 2B). The four proteins elevated in TNF are Sigma non-opioid intracellular receptor 1 (SIGMAR1, OPRS1), 60S ribosomal protein L23 (RPL23), 40S ribosomal protein S28 (RPS28) and ATP dependent RNA helicase (DDX5). Their direct or indirect role or response to TNF is currently unknown. The SIGMAR1 is an endoplasmic reticulum (ER)-resident transmembrane protein which functions in lipid transport from the ER and is involved in a wide variety of disorders, including depression, drug addiction, and pain [41]. The elevated expression of RPL23 and RPS28 might suggest that cells treated with TNF might require a more efficient translational machinery by regulating ribosome biogenesis and global protein synthesis [42]. The increased expression of DDX5 is quite interesting. DDX5 is known to participate in all aspects of RNA metabolism ranging from transcription to translation, RNA decay, and mRNA processing [43]. Its role in cell cycle regulation, tumorigenesis, apoptosis, cancer development, and adipogenesis has been well established [43]. Understanding how elevated expression of DDX5 is linked to TNF treatment will provide new information about TNF-induced cellular outcomes of adipose tissues.

Figure 3A shows the heat map of the significant proteins in each group, which showed a 2-fold change in abundances in at least one of the 3 treatments. The PCA analysis was performed to project LFQ based proteome measurements into a two-dimensional data space (Figure 3B). We applied PCA to 78 significant proteins that were quantified in each of the condition. The component 1 of the PCA (PC1) accounts for 42.6% of the variability, and clearly discriminates proteins among different treatment groups. The PC2 accounted for 20.3% of the total variations, and altogether, all 2 components accounted for 62.8% of the total variation. Proteins expressed in control segregated from the others, and proteins expressed in Ins only, TNF only, and Ins + TNF were also clearly discrete from each other. Figure 3B also shows that the distance between replicates within each group is much smaller than the separation between the groups, supporting reproducible LC-MS/MS analysis and peptide quantification, a prerequisite for accurate label-free protein quantitation

Figure 4 shows the top 5 GO biological processes and molecular functions of the 78 significantly up- or down-regulated proteins (*p* ≤ 0.05) in one or more of the 3 treatments compared to the control. For biological processes, down-regulated proteins were enriched for metabolic process, oxidation-reduction process, and TCA cycle in Ins and TNF, and ion transport and regulation and ATPase activity proteins were enriched in Ins + TNF (Figure 4A). With regard to molecular functions, down-regulated proteins were enriched for oxidoreductase activity, electron carrier activity and fatty-acyl-CoA binding activity in Ins only and TNF only treatments, and ATPase activity, voltage gated anion channel and prion activity were enriched in Ins + TNF (Figure 4C). Among the up-regulated proteins, those involved in biological processes, such as platelet degradation, translation, glycolysis, and the response to amino acids were enriched in one or more of the treatments (Figure 4B). For molecular functions, proteins involved in homodimerization, calcium ion binding, structural constituents of ribosomes, and the extracellular matrix were enriched among the up-regulated proteins (Figure 4D). The complete list of all the GO molecular functions, biological processes, and cellular components for all 693 proteins and 78 differentially expressed proteins are shown in the Appendix A. The analysis of cellular components of 693 proteins by GO annotation showed that ~30% of these proteins were resident in mitochondria (Appendix A). The same analysis showed >50% of the differentially expressed proteins (*p* ≤ 0.05) resident in mitochondria including mitochondrial matrix and mitochondrial inner membrane (Appendix A), suggesting that relatively higher mitochondrial resident proteins were affected by the treatments. Other major cellular components of differentially expressed proteins included extracellular exosomes, focal adhesion, cytoplasm/cytosol, lysosome, and cell-cell adhesion junction.

### 3.3. Differentially Expressed Mitochondrial Proteins

Of the 78 proteins (*p* ≤ 0.05), 39 proteins were resident to mitochondria (Table 1 and Appendix A). The mitochondrial creatine kinase S-type (CKMT2), which provides a spatial and temporal energy buffer to maintain cellular homeostasis, was detected only in the control samples (Appendix A). The amount of carnitine *o*-palmitoyltransferase 2 (CTP2), a mitochondrial membrane protein, decreased in all three treatments (Table 1). There was a coordinated down-regulation of proteins involved in fatty acid oxidation and tricarboxylic acid (TCA) cycle (Table 1). The expression of acyl-CoA dehydrogenases, including long-chain specific acyl-CoA dehydrogenase (ACADL), short-chain specific acyl-CoA (ACADS), very long-chain specific acyl-CoA dehydrogenase (ACADV), as well as acyl-CoA dehydrogenase family member 9 (ACAD9) and acetyl-CoA carboxylase 1 (ACACA1) were also generally down-regulated in all the treatments (Table 1 and Appendix A). These enzymes work in a stepwise fashion to fatty acid synthesis. Previous studies have shown that mice deficient in ACADL develop hepatic insulin resistance [44], and those deficient in ACADV were protected from high-fat diet-induced obesity and liver and muscle insulin resistance [45]. Inhibition of CPT2 activity is known to inhibit insulin resistance in diet-induced obese mice [46]. 

The expression of cAMP-dependent protein kinase type II (PRKAR2b), enoyl-CoA hydratase domain-containing protein 3 (ECHDC3), 3-hydroxyisobutyryl-CoA hydrolase (HIBCH), methylmalonyl-CoA mutase (MUTA) and glutaryl-CoA dehydrogenase (GCDH) also decreased in Ins and TNF treatments (Table 1, Appendix A). Several of these proteins are involved in the oxidation of unsaturated fatty acid enoyl CoA esters. Relative abundances of TCA cycle enzymes, succinate-CoA ligase (GDP-forming) subunit beta (SUCB2), and fumarate hydratase (FUMH) were also lower in Ins, TNF and both Ins + TNF. Expression of mitochondrial pyruvate dehydrogenase E1 component subunit beta (PDHB) was also reduced in all three treatments (Appendix A). PDHB catalyzes the conversion of pyruvate to acetyl-CoA, thus links the glycolytic pathway to the TCA cycle. The pyruvate dehydrogenase E1 component subunit alpha also showed reduced levels, but the alteration was not significantly different compared to the control. 

### 3.4. Glycolysis and Glycogenesis

Glycolytic enzymes triosephosphate isomerase (TPI1) and phosphoglycerate mutase 1 (PGAM1) were significantly up-regulated in all the treatments, but phosphoglycerate kinase (PGK1) was up-regulated only in Ins and Ins + TNF (Table 1). However, the expression of pyruvate carboxylase (PCX), an enzyme that plays a crucial role in lipogenesis, was not statistically significant (Appendix A). Mitochondrial glycerol-3-phosphate dehydrogenase (GPD2), an integral component of mammalian respiratory chain and glycerophosphate shuttle, was also reduced in all the treatments, as compared to the control. GPD2 is an important enzyme of intermediary metabolism that functions in between glycolysis, oxidative phosphorylation, and fatty acid metabolism [47]. Therefore, its metabolic role involves the regulation of cytosolic glycerol-3-phosphate (G3P) as a metabolite connecting glycolysis, lipogenesis, and oxidative phosphorylation. Insulin is known to reduce PEPC expression through serine/threonine kinase AKT signaling pathways [7,8]. However, in the present study, PEPC expression was not different in all the treatment groups compared to the control. Our data showed the suppression of gluconeogenesis and activation of glycogenesis in Ins, TNF and Ins + TNF, in agreement with previous reports. Mitochondrial serine/threonine-protein phosphatase 2A (PGAM5) was up-regulated under TNF treated cells (Table 1). PGAM5 displays phosphatase activity for serine/threonine residues to dephosphorylate and activate MAP3K5 kinase. This protein is known as a central mediator for programmed necrosis induced by TNF-α or by reactive oxygen species [48]. In contrast, mitochondrial inorganic pyrophosphatase 2 (PPA2) was significantly up regulated only in TNF. PPA2 hydrolyzes inorganic pyrophosphate, which is essential for correct regulation of mitochondrial membrane potential and mitochondrial membrane organization and function [49,50]. It is possible that increased mitochondrial stress and decreased mitochondrial function may cause disruption in mitochondrial potential and membrane organization, leading to increased expression of PPA2. 

The enzymes CKTM2, Slc30a9, NUDT5, NDUFA12, and NDUFC2 were down-regulated in all treated cells compared to the control (Table 1, Appendix A). The cytochrome b-c1 complex subunit 1 (UQCRC1) and cytochrome c oxidase subunit (NDUFA4) were down-regulated in all the treatments, but cytochrome c oxidase assembly factor 3 homolog (COA3) was up-regulated in Ins treated adipocytes. Similarly, β-glucuronidase (GUSB), lipid transport sigma-non-opioid intracellular receptor 1 (SIGMAR1, OPRS1) and cytoplasmic aspartate aminotransferase (GOT1) were up-regulated in all the treatments, compared to the control group. 

### 3.5. Transport, Translation, and Oxidative Stress Responses

We searched for proteins with well-established roles in structure and function of cytoskeleton, protein transport, translation, and oxidative stress responses. In contrast to the control group, the expression of collagen α-1 (I) chain (COL1a1) and collagen α-1 (IV) chain (COL4a1), which support and strengthen many tissues in the body, such as cartilage, bone, and tendon, was higher in all treatments (Table 1 and also Appendix A). The expression of 14-3-3 zeta/delta (YWHAZ) also increased in Ins, TNF, as well as Ins + TNF treatments. 14-3-3 proteins are dimeric phosphoserine-binding proteins that participate in multiple signal transduction pathways and regulate several aspects of cellular biochemistry including diabetic cardiomyopathy [51]. The Zinc transporter 9 (Slc30a9), involved in maintaining zinc concentration was detected only in the control (Appendix A). Sarcoplasmic/endoplasmic reticulum (SR) calcium ATPase 2 (ATP2a2) expression was reduced in Ins and Ins + TNF but increased in TNF only treatment (Table 1). ATP2a2 is a Ca^2+^ ATPase that transfers Ca^2+^ from the cytosol to the lumen of the SR at the expense of ATP hydrolysis during muscle relaxation [52]. The expression of calmodulin-like protein 3 (CALL3) increased in all the treated cells in comparison to the control group. 

Insulin resistance and T2D can lead to oxidative stress by elevated reactive oxygen species (ROS) generation [53]. The ROS subsequently degrade mitochondrial proteins, and key TCA cycle enzymes are particularly sensitive to such damage [54]. From our data, several distinct responses of proteins to the treatments may reflect increased ROS in mitochondria. For example, mitochondrial dihydrolipoyl dehydrogenase (DLDH) is known to serve as a source of ROS [55], and its expression increased significantly in all the treated cells (Table 1). The expression of Cu-Zn superoxide dismutase (SOD1) was not statistically significant, but generally showed higher levels in all the treatments (Figure 5, Appendix A). Mitochondrial glutathione reductase (GSR), which may play a key role in scavenging ROS was up-regulated in Ins + TNF (Figure 5, Appendix A). However, thioredixin (TRX), which also maintains cell redox homeostasis, decreased in Ins + TNF, indicating distinct responses of proteins involved in maintaining cellular redox homeostasis. Frataxin (FRDA) is a mitochondrial iron-sulfur-cluster assembly protein and affects oxidative energy flux [56]. From our data, FRDA expression was observed only in cells treated either, with TNF only or Ins + TNF. (Appendix A). This may indicate the deterioration of adipocytes due to TNF treatment. It has been shown that disruption of FRDA caused T2D, due to impaired insulin secretion and increased oxidative stress in pancreatic β cells of mice [56]. Our proteomic analysis provided further evidence that FRDA plays a role in promoting cellular defense against ROS. Our experiment may provide new insight into the deterioration of β cell function observed in different subtypes of diabetes in humans.

Mitochondrial cytochrome c oxidase, carbohydrate metabolism (β-glucuronidase), lipid transport (sigma-non-opioid intracellular receptor 1) and NADH dehydrogenase [ubiquinone] 1 alpha sub-complex assembly factor 2 (NDUFAF2) were inhibited in all the treatments groups compared to the control (Figure 3A, Table 1). USMG5 (associated protein in insulin sensitive tissues) was expressed in all except Ins + TNF. Proteins involved in vesicle transport (coatomer subunit epsilon), translation (60S and 40S ribosomal proteins), respiration (cytochrome c oxidase), carbohydrate metabolism (β-glucuronidase), lipid transport (sigma-non-opioid intracellular receptor 1), and amino acid biosynthesis were also down-regulated due to treatment with Ins, TNF, or Ins + TNF. Many of these proteins are known to exist as complexes endogenously, so future studies for the determination of their complex composition and changes due to treatment can provide new information about their structure and function.

## 4. Summary

We showed that the decreased expression of mitochondrial and cytosolic proteins involved in fatty acid beta-oxidation and carbohydrate metabolism, and increased abundances of proteins involved in folding, degradation, and stress responses. The expression of glycolytic enzymes TPI1 and PGAM1 increased in all treatments, and PGK1 increased in Ins only and Ins + TNF (Table 1). PCX, an enzyme involved in gluconeogenesis and lipogenesis showed lower expression in all the treatment groups, compared to the control (Appendix A). GS showed increased expression, but GPD2, an important enzyme of intermediary metabolism that functions in between glycolysis, oxidative phosphorylation, and fatty acid metabolism [47], was reduced in all the treatments (Appendix A). The majority of TCA cycle enzymes were particularly sensitive to the treatments. The abundance of mitochondrial DLDH, which is known to serve as a source of ROS [55], increased significantly in all the treated cells. These results showed a potential relationships among various pathways that may converge to create insulin resistance and T2D (Figure 5). What remains to be explored is how these differentially expressed proteins are organized into functional complexes and understanding how their complex composition and localization changes as a function of the Ins or TNF treatments. Additionally, the changes in post-translational modifications of these proteins due to Ins or TNF treatments will help to better understand their molecular functions and biological processes. All the raw LC-MS/MS data associated with these experiments have been deposited in the Mass Spectrometry Interactive Virtual Environment (http://massive.ucsd.edu) with the ID MSV000081520.

## Figures and Tables

**Figure 1 proteomes-07-00035-f001:**
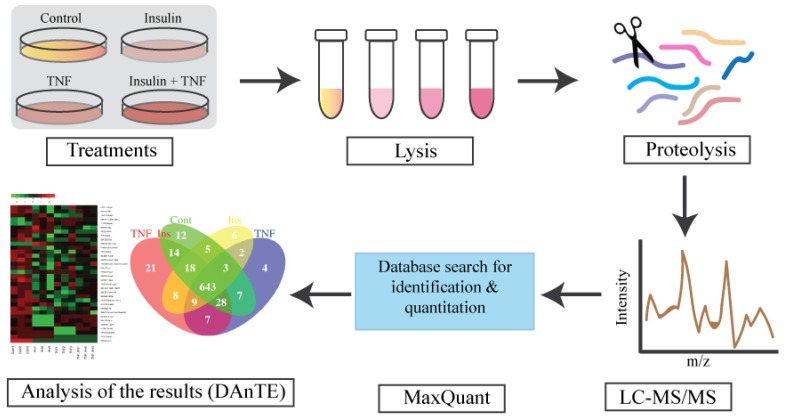
Experimental design for proteomic analysis. Fully differentiated murine 3T3-L1 adipocytes were treated with TNF-α for 8 h (TNF), insulin for 30 min (Ins), and TNFα for 8 h, followed by insulin 30 min (TNF + Ins). LC-MS/MS data were searches using MaxQuant for protein identification and quantitation. Search results were visualized and plotted using DAnTE [39]. N = 3 biological replicates from each treatment group were analyzed.

**Figure 2 proteomes-07-00035-f002:**
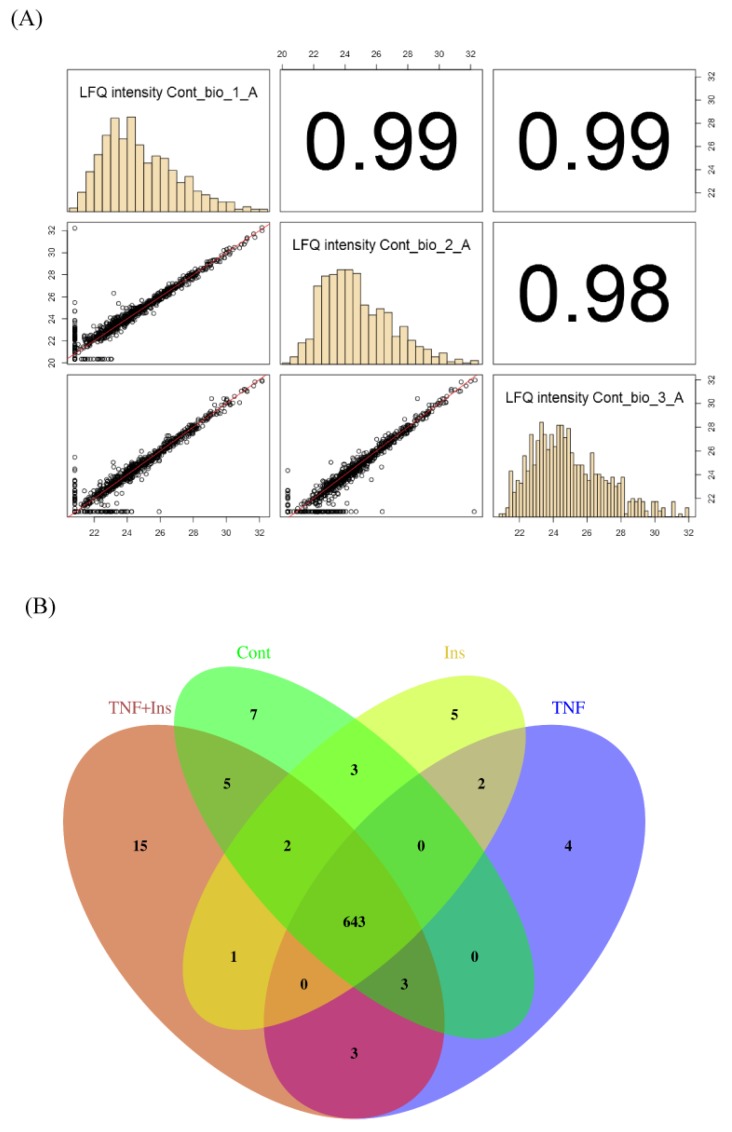
LC-MS/MS reproducibility. (**A**). Correlation plots showing reproducibility among biological replicates. (**B**) Venn diagram showing the overlap of proteins expressed across all treatments.

**Figure 3 proteomes-07-00035-f003:**
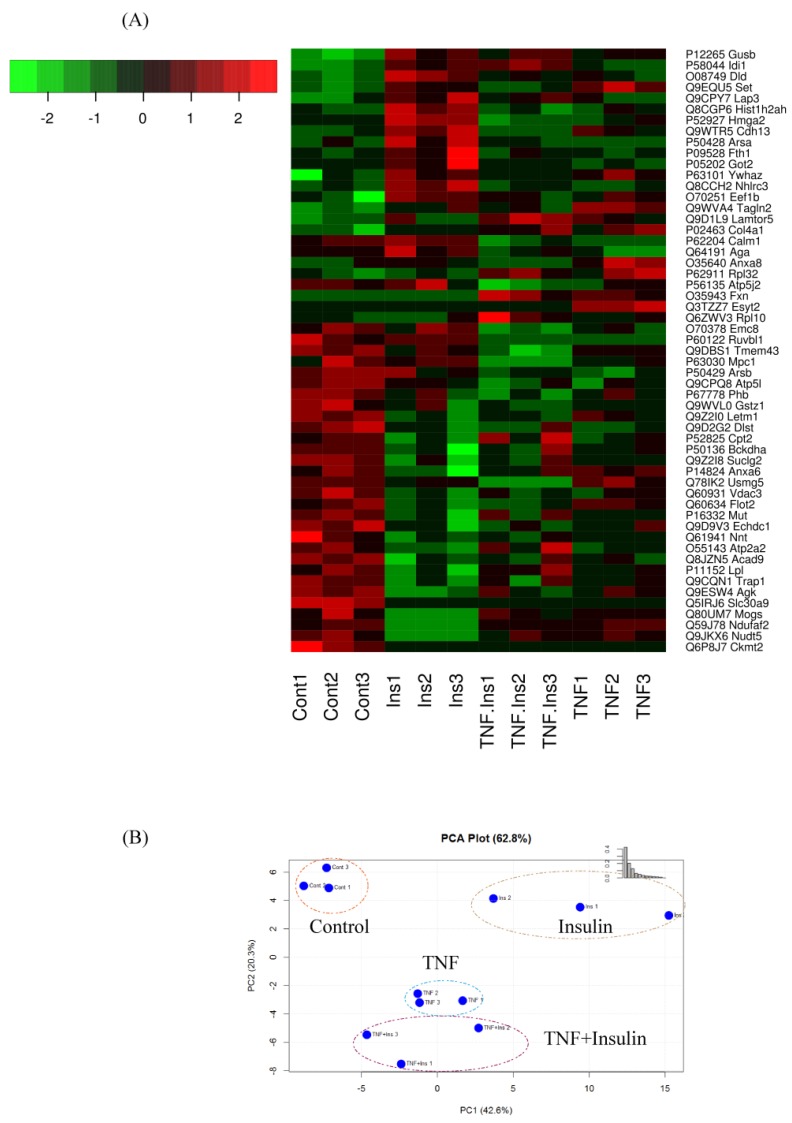
(**A**) Heatmap of differentially expressed proteins (*p* ≤ 0.05) which showed ≥ 2-fold increase or decrease in expression in one or more treatment groups compared to the control group. (**B**) Principal component analysis, using the LFQ intensity of all significant proteins, displaying the four groups of Control, Ins, TNF, and TNF + Ins.

**Figure 4 proteomes-07-00035-f004:**
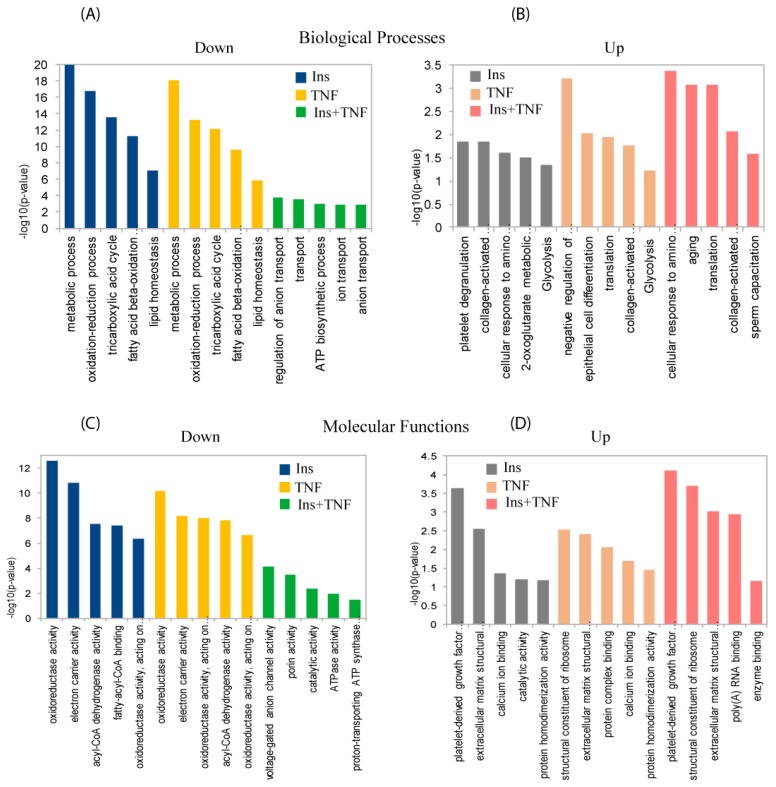
Gene ontology (GO) analysis for biological processes (upper panels) and molecular functions (lower panels) of differentially expressed proteins. Top five down-regulated (**A**) and up-regulated (**B**) biological processes in one or more treatments, compared to the control. Top five down-regulated (**C**) and up-regulated (**D**) molecular functions in one or more treatments compared to the control. The complete list of GO biological processes, molecular functions and cellular components of the total and differentially expressed proteins can be found in Appendix A, respectively.

**Figure 5 proteomes-07-00035-f005:**
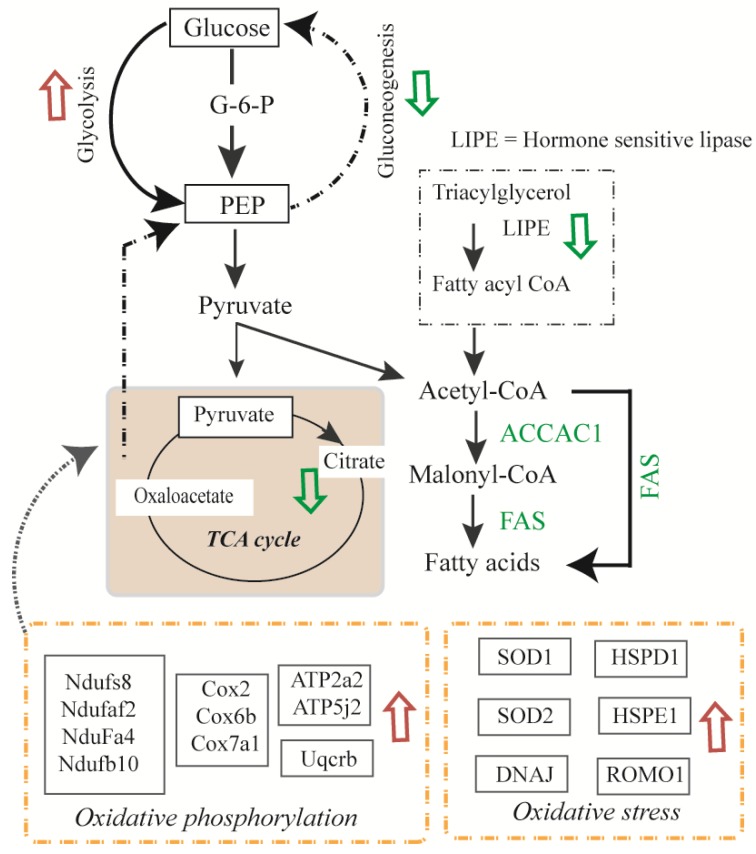
Overview of the metabolic pathway changes in murine 3T3-L1 adipocytes due to insulin or TNF-α treatments. Pathways that were up-regulated are shown by red arrows and pathways that were down-regulated are shown by green arrows. Proteins involved in pathways of glycolysis, glycogenolysis, oxidative phosphorylation and oxidative stress responses were up-regulated and protein involved in glycogenesis, gluconeogenesis, and TCA cycle, were down-regulated. LIPE; hormone sensitive lipase.

**Table 1 proteomes-07-00035-t001:** Differentially expressed proteins in cells treated with insulin (Ins), TNF-α (TNF) and insulin + TNF-α (Ins + TNF) compared to the control group. *p*-values were determined based on the ANOVA test. ‘+’ indicates up-regulation and ‘−’ indicates down-regulation of proteins in the treated cells compared to the control.

Protein ID	Protein Name	Gene	*p*-Value	Ins	TNF	TNF + Ins
Q5IRJ6	Zinc transporter 9	Slc30a9	4.0 × 10^−14^	−	−	−
Q59J78	NADH dehydrogenase [ubiquinone] 1 alpha2	Ndufaf2	1.3 × 10^−11^	−	−	−
Q78IK2	Up-regulated skeletal muscle growth protein 5	Usmg5	1.5 × 10^−10^	−	−	−
Q9JKX6	ADP-sugar pyrophosphatase	Nudt5	2.3 × 10^−8^	−	−	−
P60122	RuvB-like 1	Ruvbl1	8.0 × 10^−8^	+	−	−
Q80UM7	Mannosyl-oligosaccharide glucosidase	Mogs	8.6 × 10^−8^	−	−	−
P63030	Mitochondrial pyruvate carrier 1	Mpc1	4.3 × 10^−7^	−	−	−
P12265	Beta-glucuronidase	Gusb	0.00006079	+	+	+
P51912	Neutral amino acid transporter B(0)	Slc1a5	0.00009307	−	−	−
O08749	Dihydrolipoyl dehydrogenase, mito.	Dld	0.0001839	+	+	+
P52927	High mobility group protein HMGI-C	Hmga2	0.0002093	+	−	−
Q8CCH2	NHL repeat-containing protein 3	Nhlrc3	0.0002316	+	+	−
Q9DBS1	Transmembrane protein 43	Tmem43	0.0003764	−	−	−
Q9EQU5	Protein SET	Set	0.0004052	+	+	+
P58044	Isopentenyl-diphosphate Delta-isomerase 1	Idi1	0.0004929	+	+	+
P62204	Calmodulin (CaM)	Calm1	0.0005279	+	−	−
Q9WTR5	Cadherin-13	Cdh13	0.0008161	+	+	+
P63028	Translationally-controlled tumor protein	Tpt1	0.001126	+	+	−
O35326	Serine/arginine-rich splicing factor 5	Srsf5	0.001389	+	−	−
P02463	Collagen alpha-1(IV) chain	Col4a1	0.001912	+	+	+
O70378	ER membrane protein complex subunit 8	Emc8	0.003163	−	−	−
Q9ESW4	Acylglycerol kinase, mitochondrial	Agk	0.003542	−	−	−
O35640	Annexin A8	Anxa8	0.003733	+	−	+
Q8BX10	Serine/threonine-protein phosphatase, mito.	Pgam5	0.004051	−	−	+
Q9WVA4	Transgelin-2	Tagln2	0.004956	+	+	+
Q60634	Flotillin-2	Flot2	0.005462	−	−	−
P11087	Collagen alpha-1(I) chain	Col1a1	0.005898	+	+	+
Q9Z2I0	Mito. proton/calcium exchanger protein	Letm1	0.006284	−	−	−
Q9WVJ3	Carboxypeptidase Q	Cpq	0.006917	+	−	−
O55143	Sarcoplasmic/ER calcium ATPase 2	Atp2a2	0.008952	−	+	−
Q9D8E6	60S ribosomal protein L4	Rpl4	0.01043	−	−	+
P05202	Aspartate aminotransferase, mito.	Got2	0.01057	+	+	−
P20152	Vimentin	Vim	0.01185	+	+	+
P50428	Arylsulfatase A	Arsa	0.0131	+	+	−
P62911	60S ribosomal protein L32	Rpl32	0.01327	+	+	+
Q8BH59	Calcium-binding mito. carrier protein Aralar1	Slc25a12	0.01336	−	−	−
Q6ZWV3	60S ribosomal protein L10	Rpl10	0.01399	−	+	+
P17751	Triosephosphate isomerase	Tpi1	0.01403	+	+	+
P13020	Gelsolin	Gsn	0.01572	+	+	+
P50429	Arylsulfatase B	Arsb	0.01641	−	−	−
Q8K0D5	Elongation factor G, mito.	Gfm1	0.0175	−	−	−
P14824	Annexin A6	Anxa6	0.01751	−	−	−
Q5SWU9	Acetyl-CoA carboxylase 1	Acaca	0.01889	−	+	−
Q91VM9	Inorganic pyrophosphatase 2, mitochondrial	Ppa2	0.0189	−	−	−
Q9CZ13	Cytochrome b-c1 complex subunit 1, mito.	Uqcrc1	0.01899	−	+	+
Q9D1L9	Ragulator complex protein LAMTOR5	Lamtor5	0.01905	+	+	+
Q9WTI7	Unconventional myosin-Ic	Myo1c	0.01961	−	−	−
Q99M71	Mammalian ependymin-related protein 1	Epdr1	0.01999	+	−	−
Q8CGP6	Histone H2A type 1-H	Hist1h2ah	0.02045	+	−	+
O70251	Elongation factor 1-beta	Eef1b	0.02205	+	+	+
P27659	60S ribosomal protein L3	Rpl3	0.02291	−	+	+
Q8BGH2	Sorting & assembly component 50 homolog	Samm50	0.02302	−	−	−
Q8JZN5	Acyl-CoA dehydrogenase 9, mitochondrial	Acad9	0.02304	−	−	−
P68040	Receptor of activated protein C kinase 1	Rack1	0.02396	−	+	+
P09528	Ferritin heavy chain	Fth1	0.02567	+	−	+
Q9D2G2	Dihydrolipoyllysine succinyltransferase, mito.	Dlst	0.02608	−	−	−
Q64191	N(4)-(beta-N-acetylglucosaminyl)-L-asparaginase	Aga	0.02636	+	−	−
P67778	Prohibitin	Phb	0.02667	−	−	−
P52825	Carnitine O-palmitoyltransferase 2, mito.	Cpt2	0.02965	−	−	−
Q9CQ40	39S ribosomal protein L49, mito.	Mrpl49	0.03098	−	−	+
P29391	Ferritin light chain 1	Ftl1	0.03142	+	−	−
Q9CPY7	Cytosol aminopeptidase	Lap3	0.03145	+	+	+
P08207	Protein S100-A10	S100a10	0.03288	+	+	+
Q99KI3	ER membrane protein complex subunit 3	Emc3	0.03398	+	−	+
P56135	ATP synthase subunit f, mito.	Atp5j2	0.03428	−	−	−
Q9WTM5	RuvB-like 2	Ruvbl2	0.03593	−	−	+
P09411	Phosphoglycerate kinase 1	Pgk1	0.03644	+	−	+
Q9DBJ1	Phosphoglycerate mutase 1	Pgam1	0.03647	+	+	+
P11152	Lipoprotein lipase	Lpl	0.03691	−	−	−
Q9Z2I8	Succinate-CoA ligase beta, mito.	Suclg2	0.03695	−	−	−
P51174	Long-chain acyl-CoA dehydrogenase, mito.	Acadl	0.03766	−	-	-
Q60931	Voltage-dependent anion- channel protein 3	Vdac3	0.03796	-	−	−
Q8BJ71	Nuclear pore complex protein Nup93	Nup93	0.04064	−	−	−
P63101	14-3-3 protein zeta/delta	Ywhaz	0.04158	+	+	+
Q9CQN1	Heat shock protein 75 kDa, mito.	Trap1	0.04341	−	−	−
Q9WVL0	Maleylacetoacetate isomerase	Gstz1	0.04534	−	−	−
P31324	cAMP-dependent protein kinase type II-beta	Prkar2b	0.04755	+	+	+
P70296	Phosphatidylethanolamine-binding protein 1	Pebp1	0.04982	+	+	+

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
