# Peer review of "Proteomic Analysis of 3T3-L1 Adipocytes Treated with Insulin and TNF-α"

_proteomes, 2019, doi:10.3390/proteomes7040035_

Round 1
Reviewer 1 Report
The article titled "Proteomic analysis of insulin and TNF-a induced changes in 3T3-L1 adipocytes " by Chan et al performed proteomic analysis of TNF and insulin-treated adipocytes and found 693 proteins among which abundances of 78 proteins were significantly different.
Overall, the manuscript is well written and good. However, here are my minor comments for this manuscript.
The second sentence in the abstract is wrong. Which insulin-resistant cells contain an elevated level of circulating insulin as well as a higher amount of TNF and other cytokines. Be specific Need reference for line number 70. Are TNF or GlUT4 knockout mice are insulin resistance? Are there any human data showing that diabetic patients have a higher level of serum TNF? The authors need to explain which comes first insulin resistance or oxidative stress in TNF and insulin-treated adipocytes? What is the rationale behind the use of 10-nM insulin and insulin with TNFa? Reference is missing for page 118. In figure 2, what are those four proteins that are elevated only in TNF treated adipocytes? Elaborate on the functions of those four proteins. Please increase the font size in figure 4. It would be interesting if the authors discuss about the diabatic phenotype of major differentially regulated proteins.
Best,
Author Response
We would like to thank the reviewers and editor for taking the time to review our manuscript and make recommendations for improvement. These edits and additions provide needed clarity and help us communicate our message more effectively. We have provided both clean and files with track changes. Major addition of new text in the clean file has been highlighted in yellow.
Reviewer 1
The second sentence in the abstract is wrong. Which insulin-resistant cells contain an elevated level of circulating insulin as well as a higher amount of TNF and other cytokines. Be specific
Response: We have revised this sentence as follows:
Insulin resistant adipose tissues contain higher level of insulin than the physiological level, as well as, higher amounts of intracellular tumor necrosis factor-α (TNF-α) and other cytokines.
Need reference for line number 70.
Response: New reference [ref 18] is added to the line 70 (line 75 in the revised version). We also rephrase the sentence as: Insulin resistant individuals also contain an elevated level of circulating insulin to keep blood glucose levels under control.
Are TNF or GLUT4 knockout mice are insulin resistance?
Response: GLUT4 knockout mice are moderately insulin resistant (see ref: https://www.ncbi.nlm.nih.gov/pubmed/15044684). Male mice have hyperglycemia in the fed state, but female mice do not. Heterozygous males develop insulin-resistant diabetes, without obesity. GLUT4 knockout also develop hyperinsulinemia with hepatic insulin resistance. TNF knockout mice respond more efficiently to an exogenous dose of insulin and glucose than wild-type (see ref: https://www.ncbi.nlm.nih.gov/pubmed/10320052). This suggests that deletion of TNF-alpha leads to increased insulin sensitivity, not insulin resistance.
Are there any human data showing that diabetic patients have a higher level of serum TNF?
Response: Yes (see ref: https://www.ncbi.nlm.nih.gov/pubmed/28414180).
The authors need to explain which comes first insulin resistance or oxidative stress in TNF and insulin-treated adipocytes?
Response: We have now clearly stated by the sentence “Insulin resistance and T2D can lead to oxidative stress by elevated reactive oxygen species (ROS) generation [ref. 53] that oxidative stress is the result of mechanisms put into motion by the condition of insulin resistance (line 426-427).
What is the rationale behind the use of 10-nM insulin and insulin with TNFa? Reference is missing for page 118.
Response: The purpose of this paper is to understand how an insulin-induced high glycolytic environment impacts the adipocyte response to TNF. 10 nM insulin maximizes glucose uptake in 3T3-L1 adipocyte (http://www.jbc.org/content/290/18/11337.full)
In figure 2, what are those four proteins that are elevated only in TNF treated adipocytes? Elaborate on the functions of those four proteins.
Response: The four proteins elevated only in TNF are Sigma non-opioid intracellular receptor 1 (SIGMAR1, OPRS1), 60S ribosomal protein L23 (RPL23), 40S ribosomal protein S28 (RPS28) and ATP dependent RNA helicase (DDX5). The four proteins elevated in TNF are Signa non-opioid intracellular receptor 1 (SIGMAR1, OPRS1), 60S ribosomal protein L23 (RPL23), 40S ribosomal protein S28 (RPS28) and ATP dependent RNA helicase (DDX5). Their direct or indirect role or response to TNF is currently unknown. The SIGMAR1 is and endoplasmic reticulum (ER)-resident transmembrane protein which functions in lipid transport from the ER and is involved in a wide variety of disorders including depression, drug addiction, and pain [ref. 41]. The elevated expression of RPL23 and RPS28 might suggest that cells treated with TNF might require a more efficient translational machinery by regulating ribosome biogenesis and global protein synthesis [ref. 42]. The increased expression of DDX5 is quite interesting. DDX5 is known to participate in all aspects of RNA metabolism ranging from transcription to translation, RNA decay, and mRNA processing [ref. 43]. Its role in cell cycle regulation, tumorigenesis, apoptosis, cancer development, and adipogenesis has been well established [41]. Understanding how elevated expression of DDX5 is linked TNF treatment will provide new information about TNF-induced cellular outcomes in adipose tissues (revised manuscript line 240-286)
Please increase the font size in figure 4.
Response: We increased the font size in Figure 4. Figure 4 has been completely revised as suggested by another reviewer.
It would be interesting if the authors discuss about the diabatic phenotype of major differentially regulated proteins.
Response: We have discussed diabetic phonotypes of a few major differentially regulated proteins in the discussion (line 298-301) as follows:
Previous studies have shown that mice deficient in ACADL develop hepatic insulin resistance [44], and those deficient in ACADV were protected from high-fat diet-induced obesity and liver and muscle insulin resistance [45]. Inhibition of CPT2 activity is known to inhibit insulin resistance in diet-induced obese mice [46] (revised manuscript line 360-363).

Reviewer 2 Report
PROTEOMES-594825: In this manuscript the authors show a proteomics analysis of murine 3T·3L1 differentiated adipocytes after treatment with insulin, TNFa and both.
This manuscript is of interest; however, to my point of view, results are all mixed and not well explained making it really difficult to understand and to follow. Authors talk about decreased or augmented abundances without indicating in which situation [insulin, TNF or both]. Please note that I could not see the entire tables 1 and 2 neither had access to Supplementary Tables for revision.
Introduction:
Line 70-71: “Insulin resistant cells also contain elevated level of CIRCULATING? insulin to keep blood glucose level under control” “Consequently, conditions for insulin resistance can be modeled by treating cells with insulin or TNF-α”. These sentences make no sense; please revise. Lines 84-90: “We identified 693 proteins of which 643 proteins were common and the expression of 78 proteins was significantly different in treated cells. Results showed that mitochondrial and cytosolic proteins involved in sequential fatty acid beta oxidation and carbohydrate metabolism were down regulated [in which group?] and those involved in folding, translation and oxidative stress responses were up-regulated [in which group?]. Creatine kinase S-type (CKMT2), ADP88 sugar pyrophosphatase (NUDT5) and the NADH dehydrogenase [ubiquinone] 1 alpha sub-complex assembly factor 2 (NDUFAF2) were down-regulated [in which group?]whereas cytochrome c oxidase, beta gluronidase, and superoxide dismutase (SOD2) were up-regulated due to treatments [in which group?].”
Results and discussion.
This manuscript is really difficult to read and to follow the results. The authors show results of “all treatments” without specifying. When describing the results, there is no way to discern if treatment was insulin or TNF alone, or both. It would be very useful to better explain how many proteins (numbers) were up or down regulated for each treatment and to perform a functional analysis of those up and down regulated and also for the exclusive proteins for each group. Which were the proteins exclusively present for each treatment?
“Using a threshold of p value ≤ 0.05, we identified 78 proteins that were significantly different (Table 1)” How many were up and down regulated for each group? Where is the list of the exclusive proteins for each group?
Functional classification Figure 4: the functional analysis is completely wrong as it was performed with the 693 differentially expressed proteins without considering if they were up or down regulated; I really do not understand the meaning of this analysis. Instead, it would be more interesting to perform the same analysis with those up or down regulated proteins for each group and also comparing the different groups. The same issue is applicable to the functional analysis of the 78 proteins; I do not see the point to perform this analysis with all the proteins (up and down regulated) at once.
There is no way to follow and understand the results: example
“Of the 78 proteins (p ≤ 0.05), 39 proteins were resident to mitochondria (Table 1 and Supplementary Data Table S3). Of these 78 proteins, expression of 27 proteins decreased and 6 proteins increased in all the treatments. The mitochondrial creatine kinase S-type (CKMT2), which provides a spatial and temporal energy buffer to maintain cellular homeostasis, was detected only in the control samples (Table 2). Amount of carnitine o-palmitoyltransferase 2 (CTP2), a mitochondrial membrane protein, decreased in insulin and TNF-α (Table 1). There was a coordinated down regulation [IN WHICH SAMPLES?]of proteins involved in fatty acid oxidation and tricarboxylic acid (TCA) cycle (Table 1). Expression of acyl-CoA dehydrogenases, including long-chain specific acyl-CoA dehydrogenase (ACADL), short-chain specific acyl-CoA (ACADS), very long-chain specific acyl-CoA dehydrogenase (ACADV) as well as acyl-CoA dehydrogenase family member 9 (ACAD9) and acetyl-CoA carboxylase 1 (ACACA1) were generally down-regulated [IN WHICH SAMPLES?]
I do not understand this argument: “2-fold increase can be highly stringent for highly abundant proteins, but for low abundant proteins, this level of changes might simply represent a technical noise.” Authors show a list of proteins according to fold change, and a different one according to the p value making results very difficult to understand. I would take the statistically significant proteins with a fold change higher of 1.5.
There is no validation of results by PCR or western blot.
Author Response
We would like to thank the reviewers and editor for taking the time to review our manuscript and make recommendations for improvement. These edits and additions provide needed clarity and help us communicate our message more effectively. We have provided both clean and files with track changes. Major addition of new text in the clean file has been highlighted in yellow.
Reviewer 2
Line 70-71: “Insulin resistant cells also contain elevated level of CIRCULATING? insulin to keep blood glucose level under control” “Consequently, conditions for insulin resistance can be modeled by treating cells with insulin or TNF-α”. These sentences make no sense; please revise.
Response: This sentence has been revised as suggested and removed the word circulating
Insulin resistant adipose tissues contain higher level of insulin than the physiological level, as well as, higher amounts of intracellular tumor necrosis factor-α (TNF-α) and other cytokines (line 22-24).
Lines 84-90: “We identified 693 proteins of which 643 proteins were common and the expression of 78 proteins was significantly different in treated cells. Results showed that mitochondrial and cytosolic proteins involved in sequential fatty acid beta oxidation and carbohydrate metabolism were down regulated [in which group?] and those involved in folding, translation and oxidative stress responses were up-regulated [in which group?]. Creatine kinase S-type (CKMT2), ADP88 sugar pyrophosphatase (NUDT5) and the NADH dehydrogenase [ubiquinone] 1 alpha sub-complex assembly factor 2 (NDUFAF2) were down-regulated [in which group?]whereas cytochrome c oxidase, beta gluronidase, and superoxide dismutase (SOD2) were up-regulated due to treatments [in which group?].”
Response: We have revised and provided specific treatment information for each up- and down-regulated protein throughout the manuscript. Further, we revised Figure 4 to show up- and down-regulation of GO molecular functions and biological processes of the 78 differentially expressed protein in each treatment group.
This manuscript is really difficult to read and to follow the results. The authors show results of “all treatments” without specifying. When describing the results, there is no way to discern if treatment was insulin or TNF alone, or both. It would be very useful to better explain how many proteins (numbers) were up or down regulated for each treatment and to perform a functional analysis of those up and down regulated and also for the exclusive proteins for each group. Which were the proteins exclusively present for each treatment?
Response: We have critically revised the manuscript and have provided specific treatment information for each up- and down regulated protein throughout the manuscript. Table 1 also shows proteins up- and down-regulated in each treatment group. ‘+’ indicated up- and ‘+’ indicates down-regulated protein in Ins, TNF and Ins+TNF treatments compared to the control. We have now revised Figure 4 to show top 5 molecular functions and top biological processes of up- and down-regulated proteins under each treatment group. The manuscript has been extensively revised, and we hope this makes it more readable and easy to follow.
“Using a threshold of p value ≤ 0.05, we identified 78 proteins that were significantly different (Table 1)” How many were up and down regulated for each group? Where is the list of the exclusive proteins for each group?
Response: We have now provided the number of up- and down-regulated proteins for each group and also discussed proteins exclusively expressed in each group in the revised manuscript (line 238-256). Specifically, as suggested by another reviewer, we described the function of 4 proteins that were exclusively elevated in TNF treatment as follows:
The four proteins elevated in TNF are Sigma non-opioid intracellular receptor 1 (SIGMAR1, OPRS1), 60S ribosomal protein L23 (RPL23), 40S ribosomal protein S28 (RPS28) and ATP dependent RNA helicase (DDX5). Their direct or indirect role or response to TNF is currently unknown. The SIGMAR1 is and endoplasmic reticulum (ER)-resident transmembrane protein which functions in lipid transport from the ER and is involved in a wide variety of disorders including depression, drug addiction, and pain [41]. The elevated expression of RPL23 and RPS28 might suggest that cells treated with TNF might require a more efficient translational machinery by regulating ribosome biogenesis and global protein synthesis [42]. The increased expression of DDX5 is quite interesting. DDX5 is known to participate in all aspects of RNA metabolism ranging from transcription to translation, RNA decay, and mRNA processing [43]. Its role in cell cycle regulation, tumorigenesis, apoptosis, cancer development, and adipogenesis has been well established [43]. Understanding how elevated expression of DDX5 is linked TNF treatment will provide new information about TNF-induced cellular outcomes in adipose tissues (line 240-286).
Functional classification Figure 4: the functional analysis is completely wrong as it was performed with the 693 differentially expressed proteins without considering if they were up or down regulated; I really do not understand the meaning of this analysis. Instead, it would be more interesting to perform the same analysis with those up or down regulated proteins for each group and also comparing the different groups. The same issue is applicable to the functional analysis of the 78 proteins; I do not see the point to perform this analysis with all the proteins (up and down regulated) at once.
Response: The figure 4 does not show functional analysis of all 693 proteins. It only shows functional analysis of 78 differentially expressed proteins. We agree with the reviewers’ comments that this figure has to be modified. In the revised manuscript, we have provided a new figure which shows functional analysis of up- and down regulated proteins under each treatment group compared to the control group.
There is no way to follow and understand the results: example “Of the 78 proteins (p ≤ 0.05), 39 proteins were resident to mitochondria (Table 1 and Supplementary Data Table S3). Of these 78 proteins, expression of 27 proteins decreased and 6 proteins increased in all the treatments. The mitochondrial creatine kinase S-type (CKMT2), which provides a spatial and temporal energy buffer to maintain cellular homeostasis, was detected only in the control samples (Table 2). Amount of carnitine o-palmitoyltransferase 2 (CTP2), a mitochondrial membrane protein, decreased in insulin and TNF-α (Table 1). There was a coordinated down regulation [IN WHICH SAMPLES?] of proteins involved in fatty acid oxidation and tricarboxylic acid (TCA) cycle (Table 1). Expression of acyl-CoA dehydrogenases, including long-chain specific acyl-CoA dehydrogenase (ACADL), short-chain specific acyl-CoA (ACADS), very long-chain specific acyl-CoA dehydrogenase (ACADV) as well as acyl-CoA dehydrogenase family member 9 (ACAD9) and acetyl-CoA carboxylase 1 (ACACA1) were generally down-regulated [IN WHICH SAMPLES?]
Response: We have revised the manuscript as suggested and provided specific treatment information for each up- and down-regulated protein throughout the manuscript. Further, we revised Figure 4 to show up- and down-regulation of GO molecular functions and biological processes of the 78 differentially expressed protein in each treatment group. We also provided description of 4 proteins only detected in TNF, and also information about known T2D phenotypes of some of the differentially expressed proteins identified in this study.
I do not understand this argument: “2-fold increase can be highly stringent for highly abundant proteins, but for low abundant proteins, this level of changes might simply represent a technical noise.” Authors show a list of proteins according to fold change, and a different one according to the p value making results very difficult to understand. I would take the statistically significant proteins with a fold change higher of 1.5.
Response: We agree and have now removed that statement, and completely removed table 2. The revised manuscript focuses only those 78 proteins that we significantly differentially regulated due to the treatments.
There is no validation of results by PCR or western blot.
Response: We understand reviewer’s concern. Unfortunately, due to lack of resources, we were not able to validation experiments using Western blot or PCR. However, we argue that we applied stringent filtering criteria to confirm identified protein: 1) protein has to be identified in at least 2 out of the 3 biological replicates, 2) proteins should have a LFQ value as well as at least 2 MS/M counts for confident identification. Then, we applied rigorous statistical analysis to identify significantly different proteins.

Reviewer 3 Report
The authors describe a proteomic study of multiple changes in the 3T3L1 cell proteome in response to insulin and tnfa treatments.
The authors introduce the study in the context of insulin resistance. They then need to explain why they selected the murine 3T3L1 as a suitable model to understand resistance. Presumably the study aims to identify physiologically relevant, sustained changes in the proteome arising from chronic exposure to the selected agonists. Consequently justification of the different agonist treatment times is needed, insulin for 30min and tnfa for 8h, in terms of the study objectives and previous omic studies on 3T3L1 stimulation by insulin and/or tnfa. The Discussion section should address the likelihood of identifying increased degradation vs increased synthesis using 30min stimulation time. At line 102 a description of how the cells were harvested is required. A justification is needed for why only one protease inhibitor was used (line 105). At line 118 the missing reference should be added. In the Data Analysis section (line 148) there is no mention of removing common contaminants. If this was done it should be described. The Bonferroni correction (line 164) belongs under the Statistical Analysis method (line 166 onwards). The Results and Discussion should explain why fractionation was not employed to obtain many more identifications. Supplementary Data Tables 1-3 were missing. These must be added for review. The use of multiple pairwise t-tests (line 216 onward) is not appropriate as it increases the likelihood of false positives. This step, including Table 2, should be removed. Table 1 is partly missing as it extends beyond the page. This must be reformatted. The absence of statistical significance (lines 267 & 286) indicates no meaningful difference between values. It is therefore incorrect to infer any meaningful, underlying difference from these data. Figure 3B should be described in the text. The English needs to be corrected in multiple places throughout the text.
Author Response
We would like to thank the reviewers and editor for taking the time to review our manuscript and make recommendations for improvement. These edits and additions provide needed clarity and help us communicate our message more effectively. We have provided both clean and files with track changes. Major addition of new text in the clean file has been highlighted in yellow.
Reviewer 3
The authors introduce the study in the context of insulin resistance. They then need to explain why they selected the murine 3T3L1 as a suitable model to understand resistance.
Response: 3T3-L1 cells are an established adipocyte model. 3T3-L1 adipocyte develop insulin resistance in response to hormones (insulin), inflammatory cytokines (TNF), and saturated fatty acids. Looking at early changes, we know the transcriptional response to TNF begins within the first 1 hour of treatment, and changes in protein phosphorylation associated with insulin resistance can be detected as early as 6 hours (https://www.ncbi.nlm.nih.gov/pubmed/11978627) (line 90-94).
Presumably the study aims to identify physiologically relevant, sustained changes in the proteome arising from chronic exposure to the selected agonists. Consequently justification of the different agonist treatment times is needed, insulin for 30min and tnfa for 8h, in terms of the study objectives and previous omic studies on 3T3L1 stimulation by insulin and/or tnfa.
Response: These are not chronic time points, but represent early on changes in the proteome prior to the development of insulin resistance. The initial insulin treatment allows us to detect the impact of a high insulin/glycolytic environment on inflammatory response to TNF. The initial inflammatory response sets off a cascade of events that lead to the development of insulin resistance in these cells.
The Discussion section should address the likelihood of identifying increased degradation vs increased synthesis using 30min stimulation time. At line 102 a description of how the cells were harvested is required.
References 32 and 33 provide information on procurement and procedure for preparing the murine adipocytes used in the experiment.
A justification is needed for why only one protease inhibitor was used (line 105).
Response: PMSF (phenylmethylsulfonyl fluoride) is the most common and widely used serine protease inhibitor for any proteomics analysis. All traditional proteomic studies used only PMSF as a protease inhibitor. While other protease inhibitor cocktails are commercially available now, we thought PMSF is sufficient during the experiment. Furthermore, after lysis in the presence of PMSF, we immediately denatured proteins in the lysate using 4 volume of cold (-20°C) acetone. Therefore, we don’t think any protein degradation due to protease activity that could affect our analysis.
At line 118 the missing reference should be added.
Response: References as required have been added, as well as throughout the manuscript.
In the Data Analysis section (line 148) there is no mention of removing common contaminants. If this was done it should be described.
Response: We specify in this section, “The false discovery rate (FDR) of peptides […] The results were filtered to remove contaminants and also those identified as reverse hit. Similarly, proteins with LFQ ≠ 0 and MS/MS ≥ 2 in at least two replicates were retained for further analysis
The Bonferroni correction (line 164) belongs under the Statistical Analysis method (line 166 onwards).
Response: These sections have been merged to create a bridging between the two contexts, which should satisfy the placement of the line in question.
The Results and Discussion should explain why fractionation was not employed to obtain many more identifications. Supplementary Data Tables 1-3 were missing. These must be added for review.
Response: We did not employ fractionation because we didn’t have enough fund to cover increased instrument time for fractionated samples. The data tables have been added.
The use of multiple pairwise t-tests (line 216 onward) is not appropriate as it increases the likelihood of false positives. This step, including Table 2, should be removed.
Response: Table 2 and discussion surrounding it was removed.
The absence of statistical significance (lines 267 & 286) indicates no meaningful difference between values. It is therefore incorrect to infer any meaningful, underlying difference from these data.
Response: We have rephrased or deleted any discussion related to proteins that were not statistically significant.
Figure 3B should be described in the text. The English needs to be corrected
Response: English has been corrected throughout the manuscript and the figure has been described.

Round 2
Reviewer 2 Report
The manuscript has improved and it is much easier to follow; however, there are still minor points to be corrected.
Introduction: The alarmingly increasing prevalence of obesity and T2D is not only a United States problem; it is also a worldwide issue that needs a solution for everyone.
Results and discussion: Results are better shown now and it is now much easier to follow. However, there are certain aspects that need editing.
Lines 215-222: It is appropriate to describe the function of the 4 proteins that were exclusively elevated in TNF treatment, but also those 5 proteins present in insulin treated and especially those that were exclusively in Ins-TNF. The most relevant results should be discussed; all of them.
Line 322: A reference is missing in: “in agreement with previous reports”
Moreover, I do not see any reference to previous proteomics studies using the same cell lines highlighting the novelty of the present work:
Insulin modulates the secretion of proteins from mature 3T3-L1 adipocytes: a role for transcriptional regulation of processing. P. WangJ. KeijerA. BunschotenF. BouwmanJ. RenesE. Diabetologia October 2006, Volume 49, Issue 10, pp 2453–2462|
Comparative proteome analysis of 3T3-L1 adipocyte differentiation using iTRAQ-coupled 2D LC-MS/MS. Ye F1, Zhang H, Yang YX, Hu HD, Sze SK, Meng W, Qian J, Ren H, Yang BL, Luo MY, Wu X, Zhu W, Cai WJ, Tong JB. J Cell Biochem. 2011 Oct;112(10):3002-14. doi: 10.1002/jcb.23223.
Proteomics in the characterization of adipose dysfunction in obesity. David Brockman &Xiaoli Chen. Adipocyte Volume 1, 2012 - Issue 1
Reviewer 3 Report
Sufficient improvement has been made to the manuscript, including much improved English.